# Unraveling the Significance of EPH/Ephrin Signaling in Liver Cancer: Insights into Tumor Progression and Therapeutic Implications

**DOI:** 10.3390/cancers15133434

**Published:** 2023-06-30

**Authors:** Stavros P. Papadakos, Ioanna E. Stergiou, Nikolina Gkolemi, Konstantinos Arvanitakis, Stamatios Theocharis

**Affiliations:** 1First Department of Pathology, Medical School, National and Kapodistrian University of Athens, 11527 Athens, Greece; nikolina.gkolemi@gmail.com; 2Department of Pathophysiology, School of Medicine, National and Kapodistrian University of Athens, 11527 Athens, Greece; stergiouioa@med.uoa.gr; 3Division of Gastroenterology and Hepatology, First Department of Internal Medicine, AHEPA University Hospital, Aristotle University of Thessaloniki, St. Kiriakidi 1, 54636 Thessaloniki, Greece; arvanitak@auth.gr; 4Basic and Translational Research Unit, Special Unit for Biomedical Research and Education, School of Medicine, Faculty of Health Sciences, Aristotle University of Thessaloniki, 54636 Thessaloniki, Greece

**Keywords:** EPH/ephrin, immunotherapy, hepatocellular carcinoma, cholangiocarcinoma, targeted therapies

## Abstract

**Simple Summary:**

This narrative literature review delves into the role of EPH/ephrin signaling in liver cancer and its implications for tumor progression and potential therapeutic strategies. This review focuses on the relationship between hypoxia and HCC development, highlighting the upregulation of hypoxia-inducible factor 1-alpha (HIF-1α) in HCC cells under low oxygen conditions. It explores the significance of the EPH/ephrin axis in regulating the hypoxic tumor microenvironment (TME) of HCC. The review also highlights the new avenues for therapeutic interventions that open the targeting of the EPH/ephrin signaling pathway in liver cancer treatment by unraveling the significance of the EPH/ephrin signaling.

**Abstract:**

Liver cancer is a complex and challenging disease with limited treatment options and dismal prognosis. Understanding the underlying molecular mechanisms driving liver cancer progression and metastasis is crucial for developing effective therapeutic strategies. The EPH/ephrin system, which comprises a family of cell surface receptors and their corresponding ligands, has been implicated in the pathogenesis of HCC. This review paper aims to provide an overview of the current understanding of the role of the EPH/ephrin system in HCC. Specifically, we discuss the dysregulation of EPH/ephrin signaling in HCC and its impact on various cellular processes, including cell proliferation, migration, and invasion. Overall, the EPH/ephrin signaling system emerges as a compelling and multifaceted player in liver cancer biology. Elucidating its precise mechanisms and understanding its implications in disease progression and therapeutic responses may pave the way for novel targeted therapies and personalized treatment approaches for liver cancer patients. Further research is warranted to unravel the full potential of the EPH/ephrin system in liver cancer and its clinical translation.

## 1. Introduction

### 1.1. Epidemiology of Liver Cancer

Hepatocellular carcinoma (HCC) is the most common histologic type of liver cancer arising from the hepatocytes [1], with almost 906,000 cases diagnosed globally in 2020. It is also the third-leading cause of cancer deaths worldwide, with a low relative 5-year survival rate of approximately 18% [2]. HCC predominantly affects men and is most commonly diagnosed in people aged between 60 and 70 years [3]. The incidence of HCC varies by geographical region and ethnicity, which is largely attributed to major risk factors. Chronic liver disease resulting from chronic infections with hepatitis B virus (HBV) or hepatitis C virus (HCV), alcohol abuse, and non-alcoholic fatty liver disease (NAFLD) or non-alcoholic steatohepatitis (NASH) are the main risk factors for HCC [3,4]. Obesity, diabetes, nicotine use, hemochromatosis, and hereditary tyrosinaemia type 1 are also associated with an increased incidence of HCC [4]. The distribution of risk factors for HCC shows regional variations worldwide, with HBV being predominant in Asia, HCV in Japan, and NAFLD, NASH, and alcohol-related factors more prevalent in Europe and North America [3]. Aflatoxin B1 exposure is especially relevant in Asia, where it overlaps with HBV infection. Several scores have been established and validated to predict the remaining risk of HCC in patients with cirrhosis [5]. Antiviral treatment improves survival in patients with HBV- and HCV-related HCC [6]. Recent evidence also suggests that direct-acting antiviral agent (DAA) therapy does not increase the risk of HCC recurrence in patients with HCV-related HCC [6]. The incidence of NAFLD- and NASH-related liver cancer has increased and optimization of glycemic control and body weight are desirable as they appear to be independently associated with an increased risk of HCC [3,4,7].

Cholangiocarcinoma (CCA) is the predominant malignant tumor affecting the bile ducts and represents the second most frequent form of liver cancer, following HCC [8]. CCAs are classified as epithelial tumors that display characteristic traits associated with the differentiation of cholangiocytes [8]. CCA is accompanied by diverse molecular features that make it a promising candidate for targeted therapy [9]. In the Western world, the occurrence of CCA is quite infrequent, with a range of 0.35 to 2 instances per 100,000 individuals annually [10]. However, the global incidence of CCA has been steadily increasing over the last 30 years, from 0.1 to 0.6 cases per 100,000 people [3]. CCA is a highly aggressive cancer, with a 5-year survival rate of less than 10% for those with locally advanced or metastatic disease. CCA develops from the biliary epithelium, either intrahepatic or extrahepatic. Perihilar CCA (pCCA) accounts for 60–70% of cases, while distal CCA (dCCA) and intrahepatic CCA (iCCA) account for 20–30% and 5–10% of cases, respectively [11]. There has been an increased prevalence of CCA in recent years. The primary treatment for early-stage CCA is surgical resection with adjuvant chemotherapy, while systemic chemotherapy is the standard treatment for advanced-stage CCA. Unfortunately, patients with CCA often present with late-stage disease, which makes the prognosis poor [12].

The EPH/ephrin system plays a significant role in cancer and its dysregulation has been implicated in the development and progression of several types of cancer [13,14,15,16,17,18,19,20,21,22]. The EPH/ephrin signaling system has been identified as a potential therapeutic target in liver cancer due to its involvement in tumor growth, invasion, and metastasis. Inhibiting this system could provide a promising approach for treating liver cancer, including HCC and CCA.

### 1.2. The EPH/Ephrin Signaling System

The Erythropoietin Producing Hepatocellular (EPH) carcinoma receptor system was first described in 1987 by Hirai and colleagues [23]. This system constitutes the largest family of receptors with tyrosine kinase activity, as it consists of 16 receptors, which are divided into 2 subclasses, the type A (EPHA) and type B (EPHB) effector receptors. In total, there are 10 EPHA receptors (EPHA1 to EPHA10) and 6 EPHB receptors (EPHB1 to EPHB6). These receptors bind to their binding sites, the ephrins. By analogy with EPHs, ephrins are divided into two subgroups, the five type A ephrins (ephrinA1 to ephrinA5) and the three type B ephrins (ephrinB1 to ephrinB3) [24]. EPH tyrosine kinases and ephrins are capable of functioning as both receptors and ligands, which enables them to engage in unidirectional or bidirectional signaling. This can result in parallel or antiparallel signaling between two adjacent cells in trans configuration. Additionally, these molecules can interact with each other on the surface of the same cell in cis configuration which appears to dampen EPH/ephrin signaling, potentially by disrupting the formation of EPH clusters [25]. The activation of EPHs results in various cellular responses, including the rearrangement of the cytoskeleton, which can involve the collapse of this structure [26]. This is accomplished by modulating the activity of small GTPases, which are critical regulators of cytoskeletal dynamics. EPH forward signaling frequently leads to cell repulsion, while ephrin reverse signaling can result in either cell repulsion or adhesion [27]. The EPH/ephrin system regulates a wide range of biological processes, including cell migration and differentiation [28], angiogenesis [29], bone and placenta formation [30,31], and synaptic plasticity in the nervous system [26]. Dysregulation of the EPH/ephrin system has been implicated in a range of diseases, including cancer [13,16,18,19,20,32], cardiovascular disease [33] and neurodegenerative disorders [34].

### 1.3. The EPH/Ephrin Molecular Structure

EPHs are transmembrane type-1 proteins with a biologically conserved structure between species. The extracellular side consists of the ligand-binding domain, the cysteine-rich region (Sushi and Epidermal Growth Factor-like domains are included here), and two fibronectin domains. The intracellular part consists of the transmembrane domain, the tyrosine kinase domain, and the SAM and PDZ sequences. This structure is common between EPHA and EPHB; however, notable exceptions are EPHA10 and EPHB6, which—due to loss of relevant amino acids—lack phosphorylation capacity [35]. In contrast, type A ephrins differ from type B ephrins. Although both are localized to the cell membrane, type A are composed of the extracellular binding domain, which is in contact with the membrane via glycosylphosphatidylinositol (GPI), whereas type B are transmembrane proteins with an extracellular binding domain and an intracellular PDZ sequence [36]. This distinction of EPHs and ephrins into types A and B was made both on the basis of commonalities in terms of structure and preference in terms of acceptor, as an EPHA tends to bind to an ephrinA and vice versa, but not excluding interaction between ligand and receptor of different types [37] (Figure 1). 

## 2. The EPH/Ephrin Signaling in HCC—Preclinical Data

The EPH/ephrin signaling pathway plays a crucial role in the development and progression of HCC. Mounting evidence suggest that the aberrant activation of this pathway contributes to tumor cell proliferation, invasion, angiogenesis, and metastasis. Targeting EPH/ephrin signaling may be a promising therapeutic strategy for the treatment of HCC. Created with Biorender.com.

### 2.1. The Role of EPH/Ephrin Signaling in HCC Proliferation and Metastasis

One of the most extensively studied molecules in HCC is EPHA2. Its role is complex and context-dependent regulating HCC cell proliferation, migration, and invasion.

EPHA2: Wang et al. found that EPHA2 is crucial for tumor growth in HCC and targeting EPHA2 suppresses tumor initiation and progression, enhancing overall survival (OS) in a mouse model of HCC [38,39]. The study used CRISPR-Cas9-mediated inhibition of EPHA2 expression in the mouse liver and showed a significant reduction in tumor burden compared to the control, indicating that EPHA2 may be a potential therapeutic target for HCC. They demonstrated that EPHA2 promoted HCC development partially through activation of the AKT and STAT3 signaling pathways. The AKT signaling pathway played a critical role in promoting HCC development while STAT3 promoted stem-cell-like properties leading to tumor initiation, relapse, and drug resistance. JAKs, which are non-receptor tyrosine kinases, can directly activate STAT3 in many malignancies, including HCC [40]. JAK1 was found to have substantial expression in Huh7 and Hep3B while the expression of JAK2 was modest and JAK3 expression was very low. The study found that EPHA2 promoted STAT3 signaling through the activation of JAK1. Collectively, the application of the small molecule inhibitor ALW-II-41-27 (ALW) demonstrated significant reduction in the phosphorylation of EPHA2 and its downstream effectors in HCC cells, impairing their growth in vitro. In vivo experiments on mice with HCC xenografts showed that ALW treatment inhibited tumor growth and even caused regression [38,39]. These results suggest that targeting EPHA2 with ALW has the potential to be a therapeutic strategy for HCC. Taking this a step further, Jin et al. investigated the role of testicular nuclear receptor 4 (TR4) in HCC progression by manipulating TR4 expression in LM3 and Huh7 cells and in vivo [41]. The 3–4,5-dimethylthiazol-2-yl-5-3–carboxymethoxyphenyl-2-4-sulfophenyl-2H-tetrazolium (MTS) proliferation assay revealed little change in cell growth after altering TR4 expression, but migration and invasion abilities were significantly enhanced after knocking down TR4 and suppressed after adding TR4-cDNA. They also found that TR4 suppressed HCC cell migration and invasion via suppressing the EPHA2 expression. They showed that TR4 suppressed the expression of EPHA2 at the transcriptional level by binding to TR4-response elements (TR4REs) located on the promoter of EPHA2 [41]. It is suggested that targeting the TR4-EPHA2 signaling pathway that has been newly identified may enhance our capability to inhibit HCC metastasis. Besides the above, EPHA2 is target of epigenetic modifications [42,43]. Xiang et al. [43] aimed to investigate the role of miR-520d-3p, a tumor suppressor, and long non-coding RNA (lncRNA) myocardial infarction-associated transcript (MIAT) in HCC cells. They performed gain-of-function studies in HCC cells by transfecting miR-520d-3p mimics to overexpress miR-520d-3p. The results showed that miR-520d-3p inhibited HCC cell proliferation, promoted apoptosis, and suppressed cell migration and invasion. Moreover, miR-520d-3p inhibited the expression of VEGF and matrix metallopeptidase 9 (MMP-9), which play an important role in cell migration. They also found that the expression of MIAT was significantly elevated in both HCC tissues and cell lines. In addition, they observed an inverse relationship between the expression levels of MIAT and miR-520d-3p in HCC tissues. MIAT was found to target miR-520d-3p by bonding with its putative site in the 3′ untranslated region (3′UTR) and downregulated its expression. MIAT promoted HCC cell growth by downregulating miR-520d-3p. Finally, EPHA2 was verified as a functional target of miR-520d-3p, indicating that miR-520d-3p inhibited HCC cell proliferation by targeting EPHA2. The study suggested that miR-520d-3p and MIAT may serve as potential therapeutic targets for the treatment of HCC [43]. Finally, Niu et al. investigated the role of miR-10b-5p in suppressing the invasion and proliferation of primary hepatic carcinoma cells by downregulating EPHA2 [42]. The study demonstrated that increasing the expression of miR-10b-5p or reducing EPHA2 led to decreased cell proliferation, increased apoptosis, elevated levels of Bax and Caspase-3, and decreased levels of Bcl-2. The dual luciferase reporter assay confirmed that EPHA2 was a target of miR-10b-5p and the rescue experiment showed that transfection of pCMV-EphA2 rescued miR-10b-5p overexpression and siEphA2 rescued miR-10b-5p knockdown. The study suggests that miR-10b-5p has the potential to be a clinical target for HCC regulating the expression of EPHA2 [42].

In conclusion, EPHA2 plays a crucial role in HCC tumor growth. Inhibition of EPHA2 suppresses tumor initiation and progression, enhancing OS [40]. EPHA2 promotes HCC development through AKT and STAT3 signaling pathways. Targeting EPHA2 with ALW-II-41-27 (ALW) inhibits HCC cell growth [38,39]. TR4 suppresses HCC cell migration and invasion by suppressing EPHA2 expression [41]. miR-520d-3p and miR-10b-5p inhibit HCC cell proliferation by targeting EPHA2 [42,43]. Targeting EPHA2 and its regulatory molecules hold promise as therapeutic strategies for HCC.

EPHA1: Chen et al. [44] aimed to investigate the role of EPHA1 in angiogenesis and progression of HCC by downregulating EPHA1 using RNA interference technology. The results showed that the knockdown of EPHA1 resulted in decreased proliferation, motility, and invasion capability of HCC-derived cells in vitro. Additionally, it downregulated the expression of VEGF and MMP-2 and -9. They suggested that EPHA1 overexpression in HCC promoted cell proliferation and angiogenesis and thus EPHA1 has the potential to be a therapeutic target for HCC [44]. 

EPHA5: Yuan et al. [45] investigated the inhibitory effects of human umbilical cord-derived mesenchymal stem cells (hUCMSCs) on the proliferation and migration of HCC cells. Experimental evidence suggested that the interaction between hUCMSCs and tumor cells resulted in cell cycle arrest at specific phases and triggered apoptosis. The hUCMSC-conditioned medium attenuated the migratory abilities of the tumor cell types and downregulated the expression of Bcl-2, pro-caspase-7, β-catenin, and c-Myc, while slightly increasing the expression of EPHA5. This suggests that EPHA5 could be further investigated as biological therapy for HCC [45]. Towards the same direction, Wang et al. [46] discussed the challenges of using kinase inhibitors to target mutated driving kinases in HCC and provided evidence to demonstrate that co-activation of ALK, FGFR2, and EPHA5 serves as core kinases in HCC cells and their co-activation is required for cell growth. That is highly correlated with poor prognosis for OS. Hsp90 plays a crucial role in HCC cells by interacting with ALK, FGFR2, and EPHA5 proteins. Inhibition of Hsp90 effectively modulated the activity of these client proteins, leading to significant growth arrest. Thus, the presence of triple-positive status characterized by p-ALK, p-FGFR2, and p-EPHA5 can be viewed as a potential “combined therapeutic target” with clinical significance for treatment purposes. They suggested that Hsp90 inhibition could be an alternative method to abrogate these kinases for treatment of HCC patients. Notably, the subgroup of patients exhibiting triple-positive p-ALK/p-FGFR2/p-EphA5 markers may be particularly responsive to Hsp90 inhibitors, making them a promising target population for this therapeutic strategy [46]. 

EphrinA2: Feng et al. [47] investigated the role of ephrinA2 in HCC. EphrinA2 was found to be significantly upregulated in both HCC cell lines and clinical tissue samples, particularly in tumors invading portal veins. The overexpression of ephrinA2 in HCC cells increased their tumorigenicity in vivo, whereas knockdown of ephrinA2 had the opposite effect by inhibiting tumorigenicity. EphrinA2 was also found to confer resistance to tumor necrosis factor alpha (TNF-α)-induced apoptosis, thus promoting cancer cell survival. The study identified a novel EphrinA2/Rac1/Akt/NF-kB pathway that inhibited apoptosis in HCC cells. The findings suggest that ephrinA2 may be a potential therapeutic target for HCC [47]. 

EphrinA3: The study investigated the role of miR-210 in HCC chemotherapy with cisplatin. The study revealed that the expression of miR-210 was elevated in HCC tissues and had a positive correlation with HCC progression. Cisplatin treatment reduced the expression of miR-210, while enhancing the expression of ephrinA3, which is a target of miR-210, in HCC cells. Additionally, the overexpression of miR-210 countered the impact of cisplatin, leading to an increase in HCC cell growth, whereas inhibiting miR-210 improved the sensitivity of HCC cells to cisplatin chemotherapy. The findings suggested that miR-210-induced ephrinA3 signaling might be a potential target of cisplatin in HCC treatment [48].

EphrinA4: Pygopus-2 (Pygo2) expression is significantly higher in cancerous tissues and is associated with age, tumor size, metastasis, vascular invasion, and tumor differentiation. Patients with normal Pygo2 protein expression have longer OS and a higher 1-year survival rate than those with abnormal Pygo2 expression [49]. Yuan et al. [50] investigated the role of ephrinA4 in HCC and its regulatory mechanism. EphrinA4 was found to be highly expressed in HCC cell lines and its knockdown significantly inhibited cell proliferation, migration, and invasion in Huh7 cells. EphrinA4 was shown to interact with Pygo2 and positively regulate Pygo2 expression. In addition, the inhibition of ephrinA4 in Huh7 cells hindered the Wnt/β-catenin signaling but this effect was counteracted by PYGO2 [50]. Additionally, in vitro and in vivo studies by Lin et al. [51] showed that ephrin4 overexpression promoted HCC cell proliferation and migration, while ephrinA4 knockdown inhibited these processes. They identified that ephrinA4 directly interacted with EPHA2, leading to activation of the PIK3R2/GSK3b/b-catenin signaling pathway [51]. PIK3R2 expression was further enhanced by the overexpression of b-catenin, resulting in the establishment of a positive feedback loop. Evidence suggests that ephrinA4 is a potential therapeutic target in HCC [51]. 

EphrinA5: Wang et al. [52] investigated the regulation of ephrinA5 expression in HCC by miR-96 and miR-182. They found that the expression levels of miR-96 and miR-182 in HCC and para-tumoral liver tissues were upregulated in HCC. A reciprocal relationship was observed between the expression levels of miR-96 and miR-182 and ephrinA5 protein levels. Moreover, the direct interaction between miR-96, miR-182, and the 3′UTR region of *ephrinA5* mRNA led to the inhibition of protein translation, ultimately promoting increased proliferation and migration of HCC cells. These findings indicate that miR-96 and miR-182 function as oncomiRs in HCC by suppressing ephrinA5 expression, potentially playing significant roles in the development of hepatocarcinogenesis [52].

EphrinB2: The expression of ephrinB2 is linked to the progression of liver cancer. Dai et al. [53] examined the effects of HMQ-T-B10 (B10) on HCC, both in vitro and in vivo. They showed that B10 was able to inhibit the growth of human liver cancer cells by binding to ephrinB2 and suppressing its signaling pathway, which induced apoptosis. B10 also showed inhibitory activity on the growth of xenograft tumors derived from Hep3B in nude mice. These results suggest that B10 has potential as an effective antitumor agent for HCC [53].

### 2.2. The EPH/Ephrin Signaling in Viral Hepatitis-Related HCC

MiR-520e is downregulated in different types of cancer cells including breast cancer [54] and gastric cancer [55]. The HBx protein, one of the most important oncogenic proteins for the HBV X gene encoding, has been shown to regulate the expression of miRNAs at the transcriptional level, affecting the progression of HBV-related HCC [56]. Tian et al. [57] investigated the role of miR-520e in the growth HCC cells and the replication of HBV. They detected miR-520e and the EPHA2 in HBV-positive HCC tissues and cells. In HBV-positive HCC tissues and cells, miR-520e was shown to be upregulated and EPHA2 was downregulated. Moreover, the expression of miR-520e was found to be reduced in both Huh7-X and HepG2-X cells (normal and HCC cells that carried stable expression of HBx), but it was increased in a dose-dependent manner upon interference with HBx. The groups treated with the miR-520e mimic and si-EphA2 exhibited a decrease in the levels of EPHA2, p-p38MAPK/p38MAPK, ERK1/2, and p-ERK1/2, as well as a decrease in cell apoptosis. Conversely, these groups showed an increase in HBV-DNA content, HBsAg and HBeAg levels, and cell proliferation. Furthermore, the reversal of the promoting effect on HBV replication and tumor cell growth by the miR-520e inhibitor was achieved through the utilization of *si-ephA2*. Overall, they suggested that miR-520e plays a role in the regulation of HBV replication by inhibiting the p38MAPK and ERK1/2 signaling pathways through an inhibitory effect on EPHA2, ultimately reducing HBV replication and inhibiting tumor cell growth [57].

Regarding the HCV, Colpitts et al. [58] used RNAi screening to identify a network of kinases involved in HCV entry [59]. Although achieving sustained virological response (SVR) after direct-acting antivirals (DAAs) therapy could significantly reduce the risk of HCC [60], and SVR obtained after curative treatment for primary HCC suppressed recurrence and improved OS [61], the avoidance of HCV entry could be the ultimate preventive strategy, especially in selected high-risk populations. Lupberger et al. [59] identified a network of kinases involved in HCV entry and discovered EGFR and EPHA2 as novel co-factors. Functional experiments demonstrated that ligand-binding and kinase domains of EGFR and EPHA2 are required for HCV entry and that both RTKs are part of the same entry regulatory pathway [59]. PKIs and RTK-specific antibodies targeting EGFR and EPHA2 as HCV entry factors hold promise as a novel class of antivirals for prevention and treatment of resistant HCV infection. In fact, a clinical study aimed to assess the safety and antiviral activity of erlotinib, an oral EGFR inhibitor, in patients with chronic hepatitis C (CHC). Nine non-cirrhotic HCV patients received placebo or erlotinib (50 or 100 mg/d) for 14 days in a randomized double-blind placebo-controlled study. Erlotinib was found to be safe. During the course of the treatment, no significant reduction in HCV-RNA levels was observed. However, it is noteworthy that two out of the three patients in the erlotinib 100 mg/d group demonstrated a decrease of more than 0.5 log in HCV-RNA levels 14 days after the end of treatment (EOT), providing evidence that EGFR plays a functional role as an HCV host factor. These findings suggest that there is potential for further research on the use of erlotinib as a chemopreventive agent for hepatocellular carcinoma in patients with CHC [62].

### 2.3. The Role of EPH/Ephrin Signaling in HCC Angiogenesis

Angiogenesis is the process by which new blood vessels are formed from pre-existing ones [63,64]. It is an essential process for the growth and development of many normal tissues and organs, including the formation of blood vessels during embryonic development, wound healing, and the female reproductive cycle. However, angiogenesis can also play a critical role in the growth and spread of cancer [65,66]. There are several mechanisms that contribute to cancer angiogenesis including the release of pro-angiogenic factors such as vascular endothelial growth factor (VEGF) and platelet-derived growth factor (PDGF) from cancer cells, hypoxia, tumor-associated macrophages (TAMs), extracellular matrix remodeling, and tumor-associated endothelial cells (TECs) [65,67,68]. All of these mechanisms work together to promote the growth of new blood vessels in and around tumors, allowing them to receive the nutrients and oxygen they need to continue growing and spreading. Inhibiting angiogenesis has thus become an important target for cancer therapy [65] and mounting evidence suggests that EPH/ephrin signaling could be exploited as therapeutic target. 

*EPHA1*: EPHA1 is overexpressed in various human tumor types including HCC [44]. Recent research has shown that elevated EPHA1 expression in HCC can promote cell proliferation through stimulation by exogenous ephrinA1 [69]. In order to investigate the role of EPHA1 in HCC angiogenesis and progression, Chen et al. [44] utilized RNA interference (RNAi) technology to downregulate EPHA1 in an HCC-derived cell line with high EPHA1 expression. They established a stable knockdown clone called *SiEphA1*/Huh-7. In vitro studies demonstrated that the depletion of EPHA1 led to diminished proliferation, along with decreased motility and invasion capacity of Huh-7 cells. Moreover, siRNA-mediated *ephA1* knockdown led to downregulation of VEGF, matrix metalloproteinase (MMP)-2, and MMP-9 expression. Interestingly, suppressing EPHA1 expression in Huh-7 cells reduced their outgrowth when injected into the subcutaneous space of nude mice, potentially due to inhibition of angiogenesis, as indicated by reduced microvessel density (MVD) [44]. 

EPHA2: Wu et al. aimed to investigate the expression of hypoxia-inducible factor-2α (HIF-2α), VEGF-A, EPHA2, and MVD in residual HCC after treatment with high-intensity focused ultrasound (HIFU) ablation to assess their association with tumor recurrence and growth in HepG2 xenograft mice [70]. They found that the levels of HIF-2α, VEGFA, EPHA2, and MVD were significantly higher in residual HCC tissues than in control group tissues. The expression levels of VEGF-A and EPHA2 were strongly correlated with MVD. Additionally, there was a significant positive correlation between HIF-2α and EPHA2 expression, as well as between VEGFA and EPHA2 expression. These findings indicate a potential association between the upregulation of HIF-2α, VEGFA, and EPHA2, and angiogenesis, in the remaining HCC after HIFU ablation [70]. 

EphinA4: As mentioned above, ephrinA4 is highly expressed in patients with HCC and has an impact on the proliferation of HCC cells [50]. Yuan et al. found that *ephrinA4* knockdown impeded angiogenesis and Wnt/β-catenin signaling in HCC by downregulating PYGO2 [50]. 

EphrinB1: Sawai et al. [71] aimed to investigate the role of ephrinB1 in HCC and its possible involvement in neovascularization. They analyzed ephrinBs (B1–B3) expression in HCC and non-tumor liver tissues. They reported that the expression of ephrinB1 transcript was significantly higher in HCC tissues compared to non-tumor tissues. In vivo studies showed that HCC cells overexpressing ephrinB1 developed tumors more rapidly than control cells. The overexpressing tumors exhibited an increased number of blood vessels. Additionally, in vitro studies demonstrated that ephrinB1 induced migration and proliferation of HUVECs. They concluded that ephrinB1 may play a role in the progression of HCC by promoting neovascularization in vivo, suggesting its involvement in the development and growth of blood vessels within HCC tumors [71].

EphrinB2: Jamshidi-Parsian et al. [67] aimed to investigate the interaction between HCC cells and endothelial progenitor cells (EPCs) and its clinical significance. They utilized a co-culture system to mimic the initial interactions between tumor parenchyma (HepG2 cells) and stroma (EPC). They revealed that the paracrine interactions between HepG2 cells and EPC played a crucial role in promoting endothelial cell differentiation and angiogenesis. This effect was possibly mediated through the intercellular signaling function of exosomes released by HepG2 cells. The interaction between tumor cells and endothelial progenitor cells (EPCs) triggered enhanced migration and upregulated expression of ephrinB2 and Delta-like 4 ligand (DLL4). The level of microvesicles/exosomes in HepG2 conditioned medium (CM) was found to be inversely correlated with the levels of DLL4 and ephrinB2. As the microvesicles/exosomes were depleted from the CM, the levels of these proteins decreased accordingly. This indicated that these proteins might be secreted via exosomes, highlighting the significant role of exosomes in intercellular communication. Moreover, ephrinB2 was found to be overexpressed in HCC and cholangiocarcinoma tissue samples from human patients [67]. The above are summarized briefly in Figure 2.

### 2.4. The Role of EPH/Ephrin Signaling in Hypoxic HCC Tumor Microenvironment (TME)

In HCC, hypoxia occurs due to reduced vascularization resulting from liver injury and cirrhosis [72,73]. This hypoxia contributes to the formation of cavitary lesions in the liver as the tumor grows rapidly, leading to necrosis. Unlike normal cells, HCC cell lines exhibit normal cell cycle progression under hypoxic conditions due to the upregulation of hypoxia-inducible factor 1-alpha (HIF-1α) [74]. HIF-1α promotes the expression of growth factors such as VEGF, which stimulates tumor proliferation and hexokinases which support ATP production for HCC cells [75]. Studies have shown that elevated HIF-1α expression, measured through immunohistochemical (IHC) analysis, is associated with worse clinical outcomes in HCC patients [76]. It is closely linked to invasive characteristics of the tumor, such as capsular infiltration and portal vein invasion. Mounting evidence links the EPH/ephrin system with the regulation of hypoxic TME. Song et al. investigated the relationship between hypoxia, ephrinA1, and endothelial nitric oxide synthase (eNOS) in tumor angiogenesis [77]. Endothelial nitric oxide synthase (eNOS) is a specific type of nitric oxide synthase (NOS) enzyme expressed in endothelial cells that plays a vital role in vasodilation [78]. They found that ephrinA1, both in squamous carcinoma cells (SCC-9) and in the supernatants, was upregulated in response to hypoxic conditions. Additionally, the production of nitric oxide (NO) in HUVECs was increased during ephrinA1-induced angiogenesis. This effect was reversed when HUVECs were co-cultured with an inhibitor of eNOS called N-nitro-L-arginine methyl ester hydrochloride (L-NAME). Further analysis revealed that the phosphorylation of Akt (Ser473) and eNOS (Ser1177) was increased in HUVECs stimulated with ephrinA1, while the total expression of eNOS remained unchanged. They also demonstrated that the specific inhibitor of phosphatidylinositol 3-kinase (PI3K), called LY294002, significantly reduced the expression of phosphorylated Akt (Ser473) and phosphorylated eNOS (Ser1177) induced by ephrinA1 [77]. These findings suggest a potential mechanism by which ephrinA1 is regulated under hypoxic conditions involving a coordinated cross-talk with PI3K/Akt-dependent eNOS activation [77].

*EPHA2*: As mentioned above, sorafenib inhibits the expression of HIF-2α, VEGFA, and EPHA2 and could serve as an effective adjunct treatment for HCC following HIFU ablation, suggesting that sorafenib may help reduce the relapse rate in residual tumors after insufficient HIFU treatment [70]. EphrinA1 was found to be expressed in four HCC cell lines (PLC/PRF/5, HuH7, HepG2, and Hep3B cells) and its expression gradually increased under hypoxia in HuH7, HepG2, and Hep3B cell lines [79]. However, no increase in ephrinA1 expression was observed in PLC/PRF/5 cells. These findings indicate a correlation between hypoxic conditions and elevated ephrinA1 expression in HCC [79].

Husain et al. investigated the role of the EPHA2/ephrinA3/axis in the development of HCC and its association with hypoxia [80]. They found that ephrinA3 is upregulated by hypoxia in a HIF-1α-dependent manner and showed frequent overexpression in HCC tumors. High ephrinA3 expression in HCC tumors is associated with poorer OS, suggesting its involvement in driving poorer prognosis. In HCC cells, EPHA2 was identified as the receptor responsible for inducing self-renewal and tumor-initiating ability in response to ephrinA3 or hypoxia stimulation. The activation of the EPHA2/ephrinA3pathway upregulated the expression of ACLY, a metabolic enzyme, via SREBP1 maturation. This led to alterations in the metabolic profile of cells, including fatty acid and cholesterol synthesis and changes in intracellular ROS levels. These metabolic changes are important regulators in determining the cancer stemness of HCC cells. The presence of hypoxic niches in solid tumors, including HCC, is associated with poorer clinical features and survival outcomes [81]. Hypoxia is a major driver of intratumoral heterogeneity and has been linked to cancer stemness [82]. Husain et al. [80] demonstrated that hypoxia-induced cancer stemness in HCC is mediated by HIF-1α and involved the ephrinA3/EPHA2 axis. The EPHA2/ephrinA3axis acts as a responder to low oxygen levels, utilizes SREBP1-mediated ACLY transcription to promote metabolic reprogramming, and induces higher self-renewal and tumor-initiating capacity in HCC cells (EpCam, CD13, and CD24) [83]. These processes contribute to poorer survival outcomes in patients with HCC [84]. Targeting this pathway may have therapeutic implications for HCC treatment [80]. The above are illustrated in Figure 3. 

### 2.5. The Role of EPH/Ephrin Signaling in Epigenetic Regulation of HCC

Understanding the epigenetic alterations in HCC has provided insights into the molecular mechanisms underlying the disease and has the potential to identify diagnostic biomarkers and therapeutic targets [85]. Targeting epigenetic modifications in HCC is an active area of research and several drugs that modulate these modifications are being explored as potential therapies for HCC treatment [86]. In HCC, epigenetic alterations have been found to contribute significantly to the initiation and progression of the disease. Some of the key epigenetic mechanisms involved in HCC include DNA methylation, histone modifications, and non-coding RNA-mediated gene regulation [85]. Mounting evidence suggests that EPH/ephrin signaling comprises a target of epigenetic modification in HCC with therapeutic implications that are presented briefly in Table 1. 

## 3. The EPH/Ephrin Signaling in HCC—Clinical Importance

The EPH/ephrin system plays a significant role in HCC. Dysregulation of the EPH/ephrin system has been implicated in the development and progression of HCC, making it an important target for clinical research and potential therapeutic interventions [38,39]. 

### 3.1. The Role of EPH/Ephrin Signaling as Biomarkers

The expression levels of specific molecules of EPH/ephrin signaling have been shown to be correlated with the prognosis and clinical outcomes of HCC patients [38,46,47,50,51,80]. By incorporating these markers into clinical assessments, clinicians can better evaluate the aggressiveness of HCC and make informed treatment decisions tailored to each patient’s individual prognosis. More detailed analysis of these data is presented in Table 2. However, the implementation of such markers in clinical practice requires robust evidence from well-designed studies, including prospective trials that evaluate their predictive power and impact on treatment outcomes. Therefore, it is imperative to conduct further clinical studies to validate the prognostic significance of specific molecules within the EPH/ephrin signaling pathway in HCC [88]. 

### 3.2. The Interconnection between AFP and EPH/Ephrin Signaling in HCC

Alpha-fetoprotein (AFP) is a widely used biomarker for HCC diagnosis and monitoring. Increased levels of AFP in the serum have been consistently associated with unfavorable prognosis in HCC [89]. Elevated AFP levels have demonstrated predictive value for various clinical outcomes, including tumor recurrence after resection [90], risk of drop-out in patients awaiting liver transplantation [91], survival rates [89], response to loco-regional therapies [92], and overall survival in advanced HCC [89]. Fujiwara et al. [93] developed and validated predictive signatures based on the hepatic transcriptome and serum secretome in cohorts of 409 NAFLD patients from various regions worldwide. The prognostic liver signature (PLS)-NAFLD, consisting of a 133-gene signature, demonstrated the ability to predict the incidence of HCC over a longitudinal observation period of up to 15 years. Patients classified as high-risk based on PLS-NAFLD showed an association with IDO1+ dendritic cells, dysfunctional CD8+ T cells in fibrotic portal tracts, and impaired metabolic regulators. The integration of PLSec-NAFLD with the previously established etiology-agnostic PLSec-AFP led to enhanced categorization of HCC risk. The PLS-NAFLD signature was found to be modifiable by interventions such as bariatric surgery, lipophilic statin, and IDO1 inhibitor, suggesting its potential utility in drug discovery and as a surrogate endpoint in clinical trials for HCC chemoprevention in NAFLD [93]. 

There is emerging evidence suggesting a potential connection between AFP and the EPH/ephrin system in HCC. Iida et al. [69] investigated the significance of ephrinA1 expression in HCC, particularly in relation to AFP production. They found that ephrinA1 expression was elevated in HCC specimens and strongly correlated with AFP expression. EphrinA1 was found to induce the expression of AFP, suggesting its involvement in the mechanism of AFP induction in HCC. Additionally, ephrinA1 promoted hepatoma cell proliferation and influenced the expression of genes related to the cell cycle, angiogenesis, and cell–cell interactions. The study suggested that ephrinA1 expression contributes to the malignant characteristics of AFP-producing HCC, influencing tumor cell growth, angiogenesis, invasion, and metastasis [69]. In the same direction, Cui et al. observed a positive association between ephrinA1 expression and AFP expression in hepatoma cell lines, but an inverse association with EPHA2 expression [94]. Collectively, these findings suggest that molecules from the EPH/ephrin system (EPHA2, ephrinA1) may play a role in the pathogenesis and progression of AFP-associated HCC and could serve as biomarkers for the disease. Further clinical studies are warranted to explore the exact effectiveness of these molecules in HCC.

### 3.3. HCC Prognostic Models Taking into Consideration the EPH/Ephrin System

EPH/ephrin molecules could play a crucial role in the immunotherapy of HCC. By modulating cell–cell interactions and signaling pathways, they regulate tumor growth, invasion, and metastasis. Targeting EPH/ephrin signaling shows promise in enhancing immune responses against HCC, offering new avenues for therapeutic intervention and improved patient outcomes [95]. Huang et al. [95] conducted a clinical study aimed to investigate the prognostic and immunological significance of ephrin family genes in HCC. They investigated the association of ephrin family genes with prognosis and clinical characteristics HCC patients. They found that certain ephrin genes were differentially expressed in HCC patients. High expression of *EFNA1*, *EFNA3*, *EFNA4*, *EFNA5*, and *EFNB1* was associated with worse OS, while *EFNA2*, *EFNB2*, and *EFNB3* showed no significant correlation. *EFNA3* and *EFNA4* were linked to shorter progression-free survival (PFS) while *EFNA2* and *EFNB3* were associated with longer PFS. The expression of *EFNA1*, *EFNA3*, *EFNA4*, *EFNA5*, and *EFNB1* was negatively correlated with disease-specific survival (DSS). Multivariate analysis confirmed *EFNA3*, *EFNA4*, and *EFNB1* as independent prognostic factors for OS in HCC. Additionally, *EFNA3* and *EFNA4* were associated with advanced T stages, pathological stages, and histological grades, while *EFNA3* was also linked to vascular invasion. *EFNB1* was correlated with larger tumor size and advanced TNM stage. Age, sex, N stages, M stages and Child–Pugh grades showed no significant association with *EFNA3*, *EFNA4*, and *EFNB1* expression [95]. Furthermore, they investigated the relationship between ephrin gene expression and immune checkpoint inhibitors (ICIs) in HCC [95]. They examined the correlation between *EFNA3*, *EFNA4*, and *EFNB1* expression and immune-related biomarkers, including immune checkpoint-related genes, tumor mutational burden (TMB), and microsatellite instability (MSI). They found significant associations between ephrin gene expression and immune checkpoint-related genes. *EFNA3* and *EFNA4* showed positive correlations with *PDCD1*, *CTLA4* and *PDCD1*, *CTLA4* respectively, while *EFNB1* exhibited a strong positive correlation with *PDCD1*, *CTLA4*, *CD274*, and *PDCD1LG2*. Additionally, high TMB and MSI were associated with increased expression of *EFNA3* and *EFNA4*. Furthermore, the study explored the connection between ephrin gene expression and drug sensitivity to chemotherapy and targeted therapy. HCC patients with high *EFNA3* and *EFNA4* expression were found to be more responsive to cisplatin, doxorubicin, gemcitabine, and mitomycin C, while high *EFNB1* expression was associated with better responses to doxorubicin, gemcitabine, and mitomycin C, but poorer responses to cisplatin and sorafenib [95]. Unfortunately, they were unable to predict the response to immune checkpoint inhibitors due to the lack of available data. These findings suggest that ephrin gene expression may serve as a potential indicator for evaluating the response to chemotherapy and targeted therapy in HCC patients [95].

Several prognostic models have been developed taking into consideration various EPH/ephrin molecules [96,97,98]. The immunosuppressive nature of the TME poses a challenge for effective immunotherapy. Therefore, there is a need to identify TME-associated biomarkers for HCC [96]. Mo et al. [96] investigated the relationship between hypoxia and immunosuppression. They hypothesized that hypoxia could influence patients with immunosuppressive HCC. Patients with hypoxia displayed increased infiltration of immune cells and upregulated expression of immune checkpoint molecules. These observations suggest a potential association between hypoxia and the efficacy of immunotherapy. The infiltration of cells with immunosuppressive effects, such as monocytic lineage cells and cancer-associated fibroblasts, was more distinct in the hypoxia group, potentially aggravating immunosuppression. Although the infiltration of cells with anti-tumor immune response, such as T cells, was higher in the hypoxia group, their function might be weakened by hypoxia. The study constructed a hypoxia-associated score based on five genes (ephrinA3, dihydropyrimidinase like 4, solute carrier family 2 member 5, stanniocalcin 2, and lysyl oxidase) and they identified a hypoxia-associated subtype. The latter shows clinical potential as an independent predictive biomarker for HCC prognosis [96]. The role of ephrin signaling in HCC was investigated in a study by Yin et al. using a gene co-expression network analysis approach [97]. The objective of the study was to obtain a deeper understanding of the molecular progression of HCC and to identify groups of candidate genes linked to the gradual development of cancer. They revealed that EPH/ephrin signaling was deregulated at the very early stage of HCC and its activation increased with the progression of the disease. EPHs were found to play a role in HCC development. Specifically, EPH/ephrin signaling was associated with the suppression of apoptosis and the facilitation of cancer cell survival. The study identified several high-degree hub genes within the network modules. In a gene cluster which exhibited enrichment in cell cycle-associated genes, high-degree genes such as *GINS1*, *TOP2A*, *KIF11*, *BUB1B*, and *NEK2* were identified. Within other clusters, high-degree genes including *MUT*, *AZGP1*, *HBA1*, *HBB*, *HBD*, *HBA2*, *ACADM*, *UQCRC2*, and *SUCLA2* were discovered. These genes are associated with energy metabolism and may contribute to the observed changes in the energy source in HCC. Furthermore, the study revealed another group, which is enriched in protein ubiquitination and EPH signaling pathways, experiencing a decrease only at the initial stage of HCC. Hub genes identified in this module included *ARPC4*, *HSP90AB1*, and *ENO1* which are implicated in HCC development and related biological processes [97]. Finally, epigenetic deregulation has been identified as a significant factor in the development of human cancers. However, the specific epigenetic alterations and the potential of DNA methylation markers as prognostic biomarkers in HCC were not well understood. The study utilized tumor tissue samples obtained from 304 HCC patients who underwent surgical resection to develop a methylation-based prognostic signature [98]. The findings revealed a strong correlation between this methylation signature and well-established markers of unfavorable prognosis. Furthermore, the methylation signature maintained its independent prognostic significance for survival, along with factors such as multinodularity and platelet count. The group of patients identified by this specific pattern displayed traits associated with the molecular subtype of proliferation characterized by progenitor cell attributes. The study also confirmed the high prevalence of genes known to be deregulated by abnormal methylation in HCC (e.g., *RALGDS/AF-6*, *IGF2*, *APC*) while also identifying potential candidate epidrivers (e.g., *SEPT9* and *ephrin B2*). In summary, a validated set of 36 DNA methylation markers demonstrated the ability to accurately forecast unfavorable survival outcomes in individuals with HCC. Patients with this methylation profile displayed mRNA-based signatures indicative of tumors with progenitor cell features [98]. 

Collectively, targeting EPH/ephrin signaling shows promise in enhancing immune responses against HCC, potentially improving patient outcomes. Certain ephrin genes are associated with prognosis and clinical characteristics in HCC and their expression correlates with immune-related biomarkers and drug sensitivity. Prognostic models incorporating EPH/ephrin molecules have been developed and the identification of TME-associated biomarkers and epigenetic alterations further contributes to HCC treatment choices.

## 4. The Role of EPH/Ephrin Signaling in CCA

Molecular targeted therapy has revolutionized the treatment landscape for various malignancies in the last two decades [99,100,101,102,103,104,105]. CCA, a rare tumor with a poor prognosis, has recently seen the identification of novel molecular alterations, bringing forth the potential for targeted therapies [9,106]. In 2019, the first approved targeted therapy for locally advanced or metastatic intrahepatic CCA was pemigatinib, an inhibitor of fibroblast growth factor receptor 2 (FGFR2) gene fusions or rearrangements [107,108]. Subsequently, additional drugs targeting FGFR2 gene fusion/rearrangement received regulatory approvals as second-line or subsequent treatments for advanced CCA [109,110]. Recent approvals of tumor-agnostic therapies encompass drugs targeting mutations/rearrangements in genes such as isocitrate dehydrogenase 1 (IDH1) [111], neurotrophic tropomyosin-receptor kinase (NTRK) [112], the V600E mutation of the BRAF gene (BRAFV600E) [113], and tumors with high tumor mutational burden, high microsatellite instability, and gene mismatch repair-deficiency (TMB-H/MSI-H/dMMR) [114]. Ongoing clinical trials are exploring HER2, RET, and non-BRAFV600E mutations in CCA, along with advancements in the efficacy and safety of new targeted treatments [9]. The role of EPH/ephrin signaling in CCA is an area of active research and understanding [115,116,117,118]. 

EPHA2: EPHA2 is of paramount importance for the pathogenesis of CCA [115,116]. In their study, Xiang-Dan Cui et al. [115] conducted research to explore the impact of EPHA2 in CCA progression and metastasis, as well as the downstream signaling pathways associated with EPHA2. They revealed that EPHA2 is overexpressed in response to growth factors, leading to the activation of the mammalian target of rapamycin complex 1 (mTORC1) and extracellular signal-regulated kinase (ERK) pathways. Notably, EPHA2 activation occurred independently of ligands through phosphorylation at S897. They observed that EPHA2 overexpression promotes colony formation and facilitates tumor growth, primarily through Akt (T308)/mTORC1 activation. Additionally, abnormal EPHA2 expression and activation correlated with lower differentiation and increased metastatic potential [115]. In an orthotopic tumor model and lung metastasis model, heightened metastatic capability was observed, associated with Pyk2(Y402)/c-Src/ERK activation in addition to canonical Raf/MEK/ERK pathway activation [115]. Notably, the mTORC1 and Raf/Pyk2 pathways appeared to exert mutual influence. These findings suggest that growth factor-mediated EPHA2 plays a role in tumor growth and metastasis by activating the mTORC1 and Raf/Pyk2 pathways [115]. Towards the same direction, Yuanyuan Sheng et al. conducted a study to identify genetic aberrations during lymph node metastasis in iCCA and explore potential mechanisms and clinical strategies targeting mutations [116]. The study identified alterations in the genetic pattern associated with lymph node metastases in ICC. They found that EPHA2 was frequently mutated in ICC. Correlation analysis revealed a close association between EPHA2 mutations and lymph node metastasis in ICC. A series of in vitro and in vivo experiments demonstrated that EPHA2 mutations resulted in the phosphorylation of Ser897, independent of ligand binding, which facilitated lymphatic metastasis in ICC. They also identified the NOTCH1 signaling pathway as a significant contributor to this process. Further, they conducted in vitro assays and utilized patient-derived xenografts to show that an inhibitor specifically targeting the phosphorylation of Ser897 effectively inhibited metastasis in ICC cases with mutated EPHA2 [116]. These results strongly suggest that EPHA2 mutants hold great potential as a therapeutic target for effectively combating lymphatic metastasis in ICC. 

EPHA3/ephrinA1: Besides EPHA2, EPHA3 and ephrinA1 have been associated with metastasis in CCA [117]. Suksawat et al. investigated the levels of eNOS and phosphorylated eNOS (P-eNOS), along with their upstream regulators VEGFR3, VEGFC, EPHA3, and ephrinA1, in the Opisthorchis viverrini (Ov)/N-nitrosodimethylamine (NDMA)-induced hamster model of CCA and human CCA [117]. In human CCA, intense immunohistochemical staining of all the proteins examined was associated with metastasis. Pairwise analysis revealed a significant correlation between concurrent increases in eNOS/VEGFR3, eNOS/ephrinA1, eNOS/VEGFC, and eNOS/EPHA3 with metastasis. Moreover, elevated eNOS/VEGFR3 and eNOS/ephrinA1 were specifically associated with non-papillary type CCA. Additionally, higher levels of eNOS and P-eNOS were significantly correlated with increased microvessel density. These findings indicate that the upregulation of eNOS, P-eNOS, and their regulators is involved in the development of CCA, potentially driving angiogenesis and metastasis [117].

EPHB/ephrinB: Khansaard et al. [118] investigated the role of EPH/ephrin signaling in CCA and their association with metastasis. Immunohistochemical staining of CCA tissues from 50 patients revealed high expression of EPHB2, EPHB4, ephrinB1, and ephrinB2. Notably, high expression of EPHB2 was significantly correlated with metastatic status. Additionally, the co-expression of EPHB2/ephrinB1 and EPHB2/ephrinB2 was also significantly associated with metastasis. They further demonstrated that suppressing EPHB2 expression using siRNA reduced CCA cell migration by inhibiting the phosphorylation of focal adhesion kinase (FAK) and paxillin. These findings suggest that upregulation of EPHB2 receptors and their ligands contributes to CCA metastasis. Therefore, targeting EPHB2 expression and its downstream signaling proteins may offer potential therapeutic strategies for treating CCA.

## 5. Discussion

In patients with HCC, single-agent ICIs have been tested, resulting in objective response rates of 15–20% but without significant OS improvement [119]. Additionally, around 30% of HCC cases show inherent resistance to ICIs [120]. While promising preclinical data exist, there are still several stages that need to be successfully completed before these findings can transition into clinical practice [121,122,123]. To expand the potential benefits of immunotherapy to a wider range of patients, combination approaches have been investigated [124]. Basket trials and early phase studies have explored combinations of ICIs with anti-angiogenic agents or with other ICIs [119,125]. Encouraging results from these trials led to subsequent phase III trials evaluating the combination of anti-PD-1/PD-L1 antibodies with bevacizumab, tyrosine kinase inhibitors, or anti-CTLA-4 antibodies [126]. The IMbrave150 trial demonstrated improved survival with atezolizumab-bevacizumab, making it the first approved regimen in the front-line setting since sorafenib [127,128]. The HIMALAYA trial showed the superiority of durvalumab-tremelimumab (STRIDE regimen) over sorafenib, establishing it as a new first-line option [129]. However, combinations of ICIs and tyrosine kinase inhibitors have yielded inconsistent results, with only one phase III trial demonstrating an OS benefit [124].

The evolving therapeutic landscape for advanced HCC raises several unresolved questions that require further research [126]. These include determining optimal treatment choices and sequencing, identifying predictive biomarkers, exploring combinations with locoregional therapies, and developing new immunotherapy agents [119]. This review aims to provide an overview of the scientific rationale and available clinical data on how targeting the EPH/ephrin signaling system may revolutionize HCC management. The dysregulation of the EPH/ephrin system has been found to be involved in the development and progression of HCC, rendering it an important target for research and potential therapeutic interventions. Specific molecules within the EPH/ephrin signaling pathway have shown correlations with the prognosis and clinical outcomes of HCC patients, as shown in Table 2. Incorporating these markers into clinical assessments can help evaluate the aggressiveness of HCC and guide personalized treatment decisions. Additionally, there is evidence suggesting a potential connection between the AFP and the EPH/ephrin system in HCC [93], with certain molecules influencing AFP-associated HCC [94]. Prognostic models incorporating EPH/ephrin molecules have been developed and targeting EPH/ephrin signaling shows promise in enhancing immune responses against HCC [95,96,97,98]. These findings provide insights into potential biomarkers and therapeutic approaches for HCC. 

Personalized, targeted therapies show promise in treating CCA, as many cases have druggable mutations [130]. Future research should focus on identifying and targeting specific mutations and exploring combinations of therapies [131]. Collaboration between researchers, funding agencies, and the pharmaceutical industry is crucial for developing effective therapies and improving funding opportunities. Understanding the interactions between cancer cells, cancer stem cells, and the tumor microenvironment is essential for developing innovative treatment options, including immunotherapies [132] and ECM-oriented treatments [131]. The EPH/ephrin signaling pathway plays a significant role in CCA. Studies have shown that EPHA2, EPHA3, and ephrinA1 are involved in CCA progression and metastasis [115,116]. EPHA2, when overexpressed, activates the mTORC1 and ERK pathways independently of ligands, promoting tumor growth and metastasis [115]. EPHA2 mutations have been associated with lymph node metastasis in CCA and targeting the phosphorylation of EPHA2 at Ser897 has shown effectiveness in inhibiting metastasis [116]. EPHA3 and ephrinA1 have also been linked to metastasis in CCA. The upregulation of eNOS and its regulators, such as VEGFR3, VEGFC, EPHA3, and ephrinA1, has been implicated in angiogenesis and metastasis in CCA [117]. Additionally, high expression of EPHB2 receptors and their ligands, ephrinB1 and ephrinB2, is associated with metastasis in CCA and targeting EPHB2 expression and its downstream signaling proteins may hold therapeutic potential [118]. Overall, understanding and targeting the EPH/ephrin signaling pathway could provide new therapeutic strategies for CCA treatment.

While there is growing evidence supporting its implication in liver cancer, it is essential to understand the potential limitations of targeting this pathway for therapeutic purposes. Firstly, the available studies on EPH/ephrin in HCC and CCA suffer from limited sample sizes and inherent tumor heterogeneity, making it challenging to draw definitive conclusions. Larger and more comprehensive studies are needed to validate the observed associations and determine the clinical significance. Secondly, some studies have reported conflicting results regarding the prognostic or predictive value of EPH/ephrin molecules in HCC and CCA. These discrepancies might be attributed to differences in study design, patient cohorts, and detection methods. Further research and standardization of experimental protocols are necessary to resolve these inconsistencies. Thirdly, although preclinical studies have demonstrated the therapeutic potential of targeting EPH/ephrin signaling, the translation into clinical trials is limited. In fact, clinical trials testing its potential therapeutic applications predominantly focused on other solid malignancies [133,134,135,136]. EPHA2 has yielded controversial results. In a phase I safety and bioimaging trial, Gan et al. [133] investigated the use of DS-8895a, an afucosylated, humanized anti-EPHA2 antibody, in patients with advanced or metastatic EPHA2 positive cancers. The results indicated that DS-8895a had limited therapeutic efficacy and the biodistribution data led to the discontinuation of further development of the antibody. In contrast, Shitara et al. [134] conducted a phase I study to evaluate the safety, tolerability, pharmacokinetics, and pharmacodynamics of DS-8895a in patients with advanced solid tumors. They concluded that DS-8895a was generally well-tolerated and its exposure appeared to increase dose-dependently while inducing activated natural killer cells [134]. Analogously, EPHA2 targeted immunoliposomes were developed for sustained drug delivery to solid tumors. The liposomes were optimized with high prodrug encapsulation efficiency, stability, and a gradient of sucroseoctasulfate (SOS) for drug release. These targeted liposomes exhibited strong binding to EPHA2 and a long circulation time and demonstrated superior antitumor activity in preclinical models. The lead molecule entered a phase I clinical trial for patients with solid tumors [136].

The lack of well-designed clinical trials exploring EPH/ephrin-targeted therapies hinders the assessment of their efficacy and safety in HCC and CCA patients. Future research efforts should address these limitations to fully evaluate the potential of targeting EPH/ephrin as a therapeutic strategy for HCC and CCA patients [137]. 

## 6. Conclusions

In conclusion, the available evidence suggests that EPH/ephrin signaling plays a significant role in liver cancer, offering potential therapeutic targets for intervention. However, further research is necessary to fully elucidate the underlying molecular mechanisms, validate the clinical significance, and explore the efficacy and safety of targeting this pathway in liver cancer patients. Continued efforts in this field have the potential to pave the way for novel therapeutic strategies and improve the outcomes for individuals affected by liver cancer. 

## Figures and Tables

**Figure 1 cancers-15-03434-f001:**
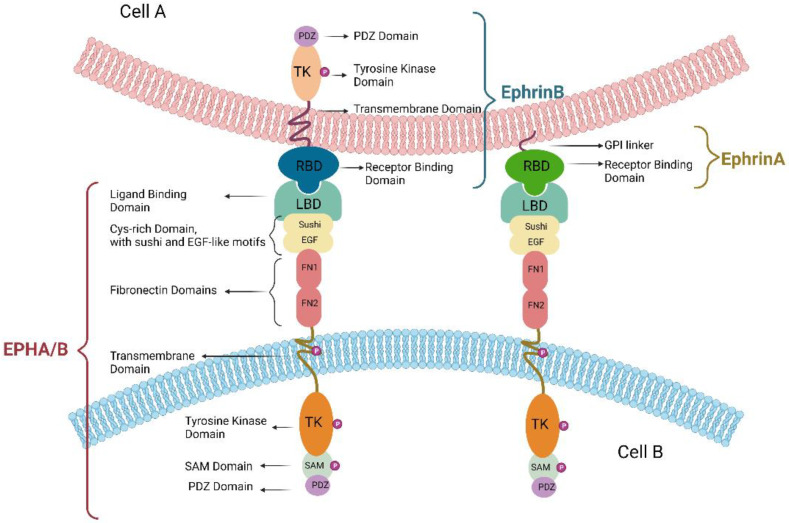
The EPH/ephrin interaction usually occurs between two adjacent cells. EPH is a transmembrane protein consisting of the extracellular domain (composed of the ligand binding domain, the cysteine-rich region, and two fibronectin domains), the transmembrane domain, and the intracellular domain (composed of the tyrosine kinase domain and the SAM and PDZ sequences). EPHA and EPHB have a common structure. In contrast, ephrinBs differ from ephrinAs. The former are transmembrane proteins with an extracellular binding domain, a transmembrane domain, and an intracellular domain with a tyrosine kinase domain and a PDZ sequence, whereas ephrinAs are synthesized from a glycosylphosphatidylinositol chain, which links the cell membrane to the binding site.

**Figure 2 cancers-15-03434-f002:**
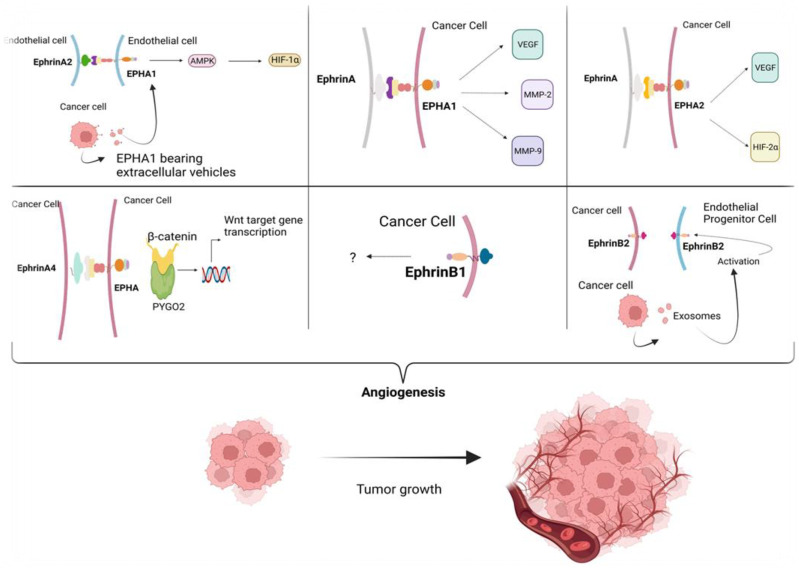
Angiogenesis plays a significant role in cancer progression. Cancer angiogenesis involves an interplay between pro-angiogenic factors, hypoxia, TAMs, ECM remodeling, and tumor-associated endothelial cells. Several research studies have investigated the role of EPHA1, ephrinA4, ephrinB1, and intercellular interactions in HCC angiogenesis. These have demonstrated that the downregulation of EPHA1 resulted in decreased HCC cell proliferation and reduced angiogenesis. Additionally, upregulation of HIF-2α, VEGF-A, EPHA2, and ephrinB1 correlated with angiogenesis in residual HCC tissues after treatment. Furthermore, interactions between HCC cells and endothelial progenitor cells promoted endothelial cell differentiation and angiogenesis through exosome-mediated signaling. Collectively, inhibiting angiogenesis has become an important target for cancer therapy and studies suggest that EPH/ephrin signaling could be a potential therapeutic target. Created with Biorender.com.

**Figure 3 cancers-15-03434-f003:**
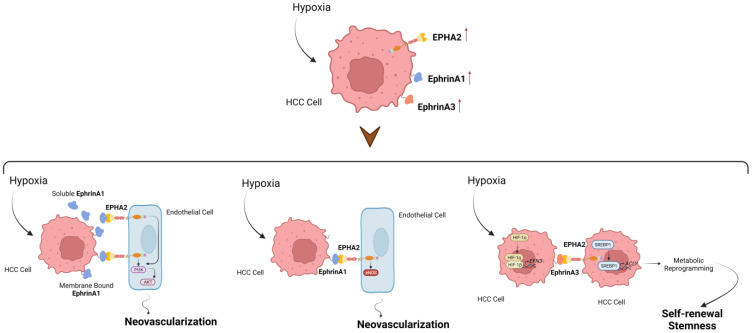
The EPH/ephrin system is also involved in regulating hypoxic conditions in HCC. EphrinA1 expression increases under hypoxia in certain HCC cell lines, while ephrinA3 is upregulated by hypoxia in a HIF-1α-dependent manner and is frequently overexpressed in HCC tumors. EphrinA3 and its receptor EPHA2 play a role in HCC development, self-renewal, and tumor initiation under hypoxic conditions. Activation of the EPHA2/ephrinA3 pathway leads to metabolic changes in HCC cells, including alterations in fatty acid and cholesterol synthesis and intracellular ROS levels, which influence cancer stemness. Hypoxic niches in HCC are associated with poorer clinical outcomes. Hypoxia-induced cancer stemness in HCC involves HIF-1α and the EPHA2/ephrinA3 axis, leading to higher self-renewal and tumor-initiating capacity. Targeting this pathway may have therapeutic implications for HCC treatment. Created with Biorender.com.

**Table 1 cancers-15-03434-t001:** Summary of the EPH/ephrin system members targeted by epigenetic modifications and their potential therapeutic implications.

Author; Year	Epigenetic Mechanism	EPH/Ephrin Target	Mechanisms	Outcomes	Ref.
Niu; 2021	miR-10b-5p	EPHA2	miR-10b-5p expression is downregulated.	miR-10b-5p plays a role in reducing cell proliferation and promoting apoptosis in HCC by regulating EPHA2. miR-10b-5p could be a promising clinical target for HCC treatment.	[42]
			EPHA2 expression is upregulated.		
			miR-10b-5p or knocking down *EphA2*: decreased cellular proliferation, facilitated apoptosis, increased expression of Bax and Caspase-3 and decreased Bcl-2.		
Xiang; 2019	miR-520d-3p	EPHA2	miR-520d-3p expression was significantly lower in HCC tissues and cells compared to tumor-adjacent tissues and normal liver cells (L02) and was associated with poor OS.	MIAT is a suppressor of miR-520d-3p and identifies EPHA2 as a direct target of miR-520d-3p with possible therapeutic implications.	[43]
			Long non-coding RNA myocardial infarction associated transcript (MIAT) was found to be upregulated in both HCC tissues and cell lines.		
			EPHA2 was identified as a direct target of miR-520d-3p and it was confirmed that MIAT functions as a competitive endogenous RNA acting as a sponge for miR-520d-3p.		
Yu; 2019	miRNA-210	ephrinA3	HCC patients who experienced tumor recurrences after chemotherapy exhibited high levels of miR-210 expression.	Targeting the miR-210-induced ephrinA3 signaling could be a potential strategy to enhance the efficacy of cisplatin-based therapies in HCC.	[48]
			Cisplatin treatment led to a decrease in miR-210 expression and an increase in ephrinA3 expression.		
			Overexpression of miR-210 counteracted the effects of cisplatin and rescued HCC cell growth, while inhibition of miR-210 improved the chemosensitivity of HCC cells to cisplatin.		
Wang; 2016	miR-96, miR-182	ephrinA5	miR-96 and miR-182 were upregulated in HCC compared to para-tumoral tissues.	miR-96 and miR-182 directly targeted ephrinA5 mRNA and suppressed its translation resulting in reduced HCC cell growth and migration.	[52]
			miR-96 and miR-182 showed an inverse relationship with ephrinA5.		
			miR-96 and miR-182 specifically bind to the 3′UTR region of *ephrinA5* mRNA.		
			Inhibition of miR-96 and miR-182 led to decreased proliferation and migration of HCC cells by negatively regulating ephrinA5 expression.		
Li; 2023	Neddylation	EPHB1	EPHB1 is neddylated by NEDD8 in HSC.	Neddylation of EPHB1 in HSCs: augmented in activated HSCs. These findings contribute to the understanding of the mechanisms underlying liver fibrosis and highlight EPHB1 as a potential target for therapeutic interventions.	[87]
			EPHB1 neddylation was enhanced by TGF-β1 stimulation and inhibited by MLN4924.		
			Neddylation was specific to EPHB1 and not in other tested EPHB family members.		

MLN4924, an inhibitor of NAE1, an enzyme involved in neddylation.

**Table 2 cancers-15-03434-t002:** Summary of the evidence about the implication of the EPH/ephrin system as a biomarker of HCC progression.

Author; Year	Molecule	Method	Outcome	Ref.
Wang; 2021	EPHA2	IHC—Y588 phosphorylated EPHA2 (p-EPHA2)—153 HCC specimens and 63 non-tumor liver tissues—The Cancer Genome Atlas (TCGA).	Increased expression of p-EPHA2 and total EPHA2 correlated with poor prognosis.	[38]
			EPHA2 signaling is correlated with a poor prognosis in HCC emphasizing its potential as a prognostic marker in this disease.	
Wang; 2019	EPHA5	Frozen tissue HCC patients.	Abnormal activation of ALK, FGFR2 and EPHA5 in a subset of HCC patients.	[46]
			The concurrent activation of ALK, FGFR2 and EPHA5 could serve as a stratifying factor to identify a subgroup of HCC patients with an unfavorable prognosis.	
			This subgroup may benefit from targeted therapeutic interventions, highlighting the potential for personalized treatment approaches.	
Hussain; 2022	ephrinA3	TCGA-LIHC, HKU-QMH cohorts.	Over two-fold overexpression of ephrinA3	[80]
			Increased expression of EFNA3 (>−4-fold) was associated with a more aggressive phenotype of HCC (venous invasion and more advanced TNM stage).	
			Higher ephrinA3 expression had poorer OS in the TCGA-LIHC cohort.	
Feng; 2010	ephrinA2	52 pairs of liver tissue: hcc vs. non-cancerous tissue.	EphrinA2 was lowest in normal liver tissues, relatively higher in primary HCCs and further elevated in portal vein tumor thrombus(PVT).	[47]
			This observation suggests that ephrinA2 plays a role as prognostic biomarker for PVT.	
Lin; 2021	ephrinA4	IHC	EphrinA4 expression was significantly higher in liver tumor tissue compared to adjacent tissue.	[51]
			A correlation between EFNA4 expression and AFP, as well as the risk of vascular invasion.	
Yuan; 2022	ephrinA4	GEPIA database	Upregulation of ephrinA4 expression in tumor samples of HCC patients compared to normal samples.	[50]
			Correlated with the TNM stages of the patients.	
			High level of ephrinA4 expression was positively associated with reduced OS and DFS.	

AFP, alpha-fetoprotein.

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
