# Peer review of "Unraveling the Significance of EPH/Ephrin Signaling in Liver Cancer: Insights into Tumor Progression and Therapeutic Implications"

_cancers, 2023, doi:10.3390/cancers15133434_

Round 1
Reviewer 1 Report
The manuscript on "Unraveling the Significance of EPH/ephrin Signaling in Liver Cancer: Insights into Tumor Progression and Therapeutic Im plications" by Papadakos et al., is a very detailed and comprehensive review on the role of EPH/ephrin signaling in liver cancer. Definitely, this review will help readers to learn the role of this very important molecule in the liver cancer. There is no major issue in the manuscript, however, following points need to correct.
1) Line 19, instead of this study, add "this review".
2) What is CCA, describe this abbreviation in line 66.
3) The connection is missing between first and second paragraph of introduction.
4) Define the abbreviation MTS in line 165.
5) Be consistent in using in vivo and in vitro; either in italics or not.
6) Add "al" after the Sawai et in line 476.
Author Response
First Department of Pathology
Laikon General Hospital of Athens
National and Kapodistrian University of Athens
June 25th, 2023
Dear Editor,
RE: Unraveling the Significance of EPH/ephrin Signaling in Liver Cancer: Insights into Tumor Progression and Therapeutic Implications
We thank you and the Reviewers for carefully evaluating our manuscript and for their positive and constructive feedback. Hopefully, our responses below address the points raised by the reviewers. All changes made are presented in the revised manuscript via the “track changes” option in MS word.
Reviewer 1: The manuscript on "Unraveling the Significance of EPH/ephrin Signaling in Liver Cancer: Insights into Tumor Progression and Therapeutic Im plications" by Papadakos et al., is a very detailed and comprehensive review on the role of EPH/ephrin signaling in liver cancer. Definitely, this review will help readers to learn the role of this very important molecule in the liver cancer. There is no major issue in the manuscript, however, following points need to correct.
Response: Dear Reviewer,
Thank you for taking the time to review our article and for providing us with your insightful feedback. We appreciate your positive comments regarding the comprehensive review of the current literature on EPH/ephrin signaling in liver and its potential use as an therapeutic target for HCC and Cholangiocarcinoma.
Point 1. Line 19, instead of this study, add "this review".
Response: We thank the reviewer for the comment.
Point 2. What is CCA, describe this abbreviation in line 66.
Response: We thank the reviewer for the comment. We have revised the text accordingly.
Point 3. The connection is missing between first and second paragraph of introduction.
Response: Dear Reviewer, thank you for your valuable feedback and the opportunity to improve our paper. We have provided additional information about the differences in histologic origin between HCC and CCA discussed in the first and second paragraphs. By highlighting these distinctions, We aimed to strengthen the overall coherence and ensure a seamless transition between the two sections.
Point 4. Define the abbreviation MTS in line 165
Response: Dear Reviewer, thank you for your valuable feedback. We have revised accordingly.
Point 5. Be consistent in using in vivo and in vitro; either in italics or not.
Response: Dear Reviewer, thank you for your valuable feedback. We have revised accordingly. We used italics.
Point 6. Add "al" after the Sawai et in line 476.
Response: Dear Reviewer, thank you for your valuable feedback. We have revised accordingly.
Reviewer 2 Report
This review manuscript by Papadakos et al focuses on the Eph family of receptor tyrosine kinases and their ephrin ligands and their roles in proliferation, metastasis, angiogenesis, metabolism, epigenetic regulation of liver cancer. They also discuss potential of Eph/ephrins as biomarkers in this disease. Overall, the authors provide a good overview of the different Eph/ephrin family members and the variety of upstream and downstream mechanisms that they are involved in to regulate hepatocellular carcinoma (HCC), and more specifically cholangiocarcinoma. One of the weaknesses of the review is the organization. It often reads as a list of results from the literature, without much explanation. Currently, in the field, there was a recent review on Eph and Ephrins in a variety of hepatic diseases, including haptic carcinoma (Mekala S et al. Ephrin-Eph receptor tyrosine kinases for potential therapeutics against hepatic pathologies. J Cell Commun Signal. 2023.), but there is a lack of reviews specifically focused on Eph and Ephrins in hepatocellular carcinoma. Because of this, I think this review would be of interest to the Eph/Ephrin and cancer community.
General Comments:
· The authors provide a nice overview of HCC in the introduction, but then they quickly jump to CCA. From just reading the review, it is unclear what CCA is and how it relates to HCC. Some additional explanation in the introduction would be helpful. It would also be beneficial to include a discussion of similarities and/or differences of Eph/Ephrin expression in HCC and CCA.
· The authors often conclude each section noting that a certain Eph receptor/protein/miR has the potential to be a clinical target or therapeutic strategy for HCC. It would be useful to expand on this idea and provide possible examples of these targets being used in the clinic, possibly for other types of cancers.
· In section 2.1 the authors breakdown the sections by specific Eph receptor or Ephrin, but then in the remaining sections they explain each topic in paragraph form. I think it would be beneficial to follow the same organization pattern in each section. It would most likely be best to edit section 2.1 to make it into paragraph form. Especially, for the EphA2 section in 2.1.1, where many of the different key points are hard to discriminate based on its current format.
· When discussing the mechanism of TR4, miR-520d-3p/MIAT, and miR-10b-5p in section 2.1.1 the focus often seems to be on these factors rather than on EphA2. Each section starts off with a thorough summary of how the pathway affects liver cancer and then briefly mentions its connection to EphA2 at the end. With this organization, the mechanistic role of EphA2 in these pathways almost seems to be an afterthought. Since the focus of the review is on the Eph receptors, it would be valuable to reorganize these sections to draw more attention to the role of EphA2 in each of these pathways.
· There seem to be some discrepancies in the miR-10b-5p section of the EphA2 subsection in 2.1.1. Wouldn’t a luciferase assay confirm that EphA2 is a target of miR-10b-5p instead of confirming that “miR-10b-5p was a target of EphA2”? The explanation of the rescue is also confusing. It would be beneficial to specifiy that pCMV-EphA2 rescues miR-10b-5p overexpression and siEphA2 rescues miR-10b-5p knockdown.
· In the EphA5 subsection in 2.1.1 the authors mentioned that Hsp90 inhibition could be an alternative treatment for HCC patients. However, there was no prior mention of Hsp90 and how it connects to EphA5.
· I believe there is a reference missing for the EphrinA3 subsection in 2.1.1. The section and the relationship between EphrinA3, cisplatin, and miR-210 is also difficult to understand. Reorganization or additional details would be helpful.
· When discussing Erlotinib (line 307) they mention that it was “found to be safe which suggests that EGFR has a functional role as an HCV host factor in patients”. However, safety of a drug in a patient does not show a functional role of EGFR.
· In the EphrinB2 panel of figure 2, the authors show that exosomes cause EphrinB2 activation in endothelial progenitor cells. The authors mention that exosomes cause an upregulation of EphrinB2, but there is lack of an explanation of how these exosomes cause EphrinB2 activation and what the downstream signaling events and outcomes of this activation may be.
· In the discussion of EphrinA1/EphA2 and hypoxia the authors discuss that activation of eNOS is downstream of EphrinA1 stimulation. It would be beneficial to add this downstream component to the Neovascularization panel in Figure 3.
· In lines 578-584 it is odd that the readers are being referred to colored sections/modules from the primary article.
· The authors introduce eNOS in lines 659-660, but eNOS was already explained previously in section 2.3. It would be useful to move this introduction to that earlier section.
Specific Comments:
· In the figure 1 legend, the authors wrote “binder” instead of “binding” a couple of times.
· The connection between EphA2, AKT, and STAT3 in lines 151-153 is repetitive with lines 148-149.
· Reference 41 is missing authors in the References section.
· The only time the abbreviation for AFP is written out is at the bottom of Table 2. It would be helpful to the reader to also include Alpha-fetoprotein written out prior to its abbreviation in line 485.
· Beginning in line 645, what is iICC?
There were minor errors in English language that I think could easily be corrected during the review process.
Author Response
Reviewer 2: This review manuscript by Papadakos et al focuses on the Eph family of receptor tyrosine kinases and their ephrin ligands and their roles in proliferation, metastasis, angiogenesis, metabolism, epigenetic regulation of liver cancer. They also discuss potential of Eph/ephrins as biomarkers in this disease. Overall, the authors provide a good overview of the different Eph/ephrin family members and the variety of upstream and downstream mechanisms that they are involved in to regulate hepatocellular carcinoma (HCC), and more specifically cholangiocarcinoma. One of the weaknesses of the review is the organization. It often reads as a list of results from the literature, without much explanation. Currently, in the field, there was a recent review on Eph and Ephrins in a variety of hepatic diseases, including haptic carcinoma (Mekala S et al. Ephrin-Eph receptor tyrosine kinases for potential therapeutics against hepatic pathologies. J Cell Commun Signal. 2023.), but there is a lack of reviews specifically focused on Eph and Ephrins in hepatocellular carcinoma. Because of this, I think this review would be of interest to the Eph/Ephrin and cancer community.
Response: Dear Reviewer,
Thank you sincerely for your thoughtful and constructive review of our manuscript. We greatly appreciate your positive feedback and recognition of the significance of our work to the EPH/Ephrin and cancer research community. We acknowledge your comment regarding the organization of the manuscript and we agree that it is crucial to provide a clear and cohesive narrative for the readers. We have taken your feedback into consideration and have made significant revisions to improve the overall structure and flow of the review.
We genuinely appreciate your recognition of the importance of our review to the EPH/Ephrin and cancer community. Your feedback has encouraged us to further enhance the clarity and relevance of our manuscript and we are committed to addressing all of your suggestions.
Once again, we extend our gratitude for your time and expertise in reviewing our work. If you have any further comments or recommendations, we would be more than happy to consider them.
Point 1: The authors provide a nice overview of HCC in the introduction, but then they quickly jump to CCA. From just reading the review, it is unclear what CCA is and how it relates to HCC. Some additional explanation in the introduction would be helpful. It would also be beneficial to include a discussion of similarities and/or differences of Eph/Ephrin expression in HCC and CCA.
Response: Dear Reviewer, thank you for your valuable feedback and the opportunity to improve our paper. We have provided additional information about the differences in histologic origin between HCC and CCA discussed in the first and second paragraphs. By highlighting these distinctions, we aimed to strengthen the overall coherence and ensure a seamless transition between the two sections.
Point 2: The authors often conclude each section noting that a certain Eph receptor/protein/miR has the potential to be a clinical target or therapeutic strategy for HCC. It would be useful to expand on this idea and provide possible examples of these targets being used in the clinic, possibly for other types of cancers.
Response: Dear Reviewer, thank you for your constructive comment. I agree that expanding on the potential clinical targets and providing examples of their use in the clinic would be valuable. It would enhance the understanding of the practical implications and therapeutic strategies involving EPHs/ephrins, not only for HCC but also for other types of cancers. This additional information could shed light on the translational potential of the findings and their application in clinical settings. Towards this direction, we extended the paragraph in Discussion which refers to the limitations of our review with data from Phase I clinical studies targeting EPH/ephrin pathway molecules.
Point 3: In section 2.1 the authors breakdown the sections by specific Eph receptor or Ephrin, but then in the remaining sections they explain each topic in paragraph form. I think it would be beneficial to follow the same organization pattern in each section. It would most likely be best to edit section 2.1 to make it into paragraph form. Especially, for the EphA2 section in 2.1.1, where many of the different key points are hard to discriminate based on its current format.
Response: Dear Reviewer, Thank you for your feedback. We have revised the organization of the sections to maintain consistency throughout the article. Section 2.1 has been edited to follow a paragraph format, similar to the remaining sections. Specifically, we have made improvements to the EphA2 section in 2.1.1 to enhance the clarity and distinguishability of the key points. We appreciate your suggestion and believe that these revisions have improved the overall readability and flow of the manuscript.
Point 4: When discussing the mechanism of TR4, miR-520d-3p/MIAT, and miR-10b-5p in section 2.1.1 the focus often seems to be on these factors rather than on EphA2. Each section starts off with a thorough summary of how the pathway affects liver cancer and then briefly mentions its connection to EphA2 at the end. With this organization, the mechanistic role of EphA2 in these pathways almost seems to be an afterthought. Since the focus of the review is on the Eph receptors, it would be valuable to reorganize these sections to draw more attention to the role of EphA2 in each of these pathways.
Response: Dear Reviewer,
Thank you for your insightful comments. We agree with your assessment regarding the organization and emphasis of the sections discussing the mechanism of TR4, miR-520d-3p/MIAT and miR-10b-5p in relation to EPHA2. In our revised version, we have reorganized these sections to highlight the role of EPHA2 more prominently by providing a thorough summary of how each pathway affects liver cancer and then emphasizing the connection to EPHA2 throughout the section, we aimed to better clarify the role of EPHA2 in these pathways.
Thank you for your valuable feedback, which has greatly contributed to the improvement of our manuscript.
Point 5: There seem to be some discrepancies in the miR-10b-5p section of the EphA2 subsection in 2.1.1. Wouldn’t a luciferase assay confirm that EphA2 is a target of miR-10b-5p instead of confirming that “miR-10b-5p was a target of EphA2”? The explanation of the rescue is also confusing. It would be beneficial to specifiy that pCMV-EphA2 rescues miR-10b-5p overexpression and siEphA2 rescues miR-10b-5p knockdown.
Response: Dear Reviewer,
Thank you for your feedback. We appreciate your keen observation regarding the discrepancies in the miR-10b-5p section of the EPHA2 subsection (2.1). You are correct and we apologize for the confusion in our previous version. In our revised manuscript, we have made the necessary corrections. We now state that the luciferase assay confirms EPHA2 as a target of miR-10b-5p, rather than the other way around. Additionally, we have provided a clearer explanation of the rescue experiments, specifying that pCMV-EphA2 rescues miR-10b-5p overexpression, and siEphA2 rescues miR-10b-5p knockdown.
Thank you for bringing these discrepancies to our attention. Your input has helped improve the accuracy and clarity of our manuscript.
Point 6: In the EphA5 subsection in 2.1.1 the authors mentioned that Hsp90 inhibition could be an alternative treatment for HCC patients. However, there was no prior mention of Hsp90 and how it connects to EPHA5.
Response: Dear Reviewer,
Thank you for pointing out the oversight. We appreciate your feedback. We have revised the manuscript accordingly to include the connection between Hsp90 and EPHA5, clarifying that Hsp90 inhibition could be a potential alternative treatment for HCC patients. The revised section now provides a comprehensive understanding of the relationship between Hsp90 and EPHA5 in the context of HCC.
Point 7: I believe there is a reference missing for the EphrinA3 subsection in 2.1.1. The section and the relationship between EphrinA3, cisplatin, and miR-210 is also difficult to understand. Reorganization or additional details would be helpful.
Response: Dear Reviewer, thank you for your valuable feedback. We have revised accordingly.
Point 8: When discussing Erlotinib (line 307) they mention that it was “found to be safe which suggests that EGFR has a functional role as an HCV host factor in patients”. However, safety of a drug in a patient does not show a functional role of EGFR.
Response: Dear Reviewer, thank you for your valuable feedback. We have revised accordingly.
Point 9: In the EphrinB2 panel of figure 2, the authors show that exosomes cause EphrinB2 activation in endothelial progenitor cells. The authors mention that exosomes cause an upregulation of EphrinB2, but there is lack of an explanation of how these exosomes cause EphrinB2 activation and what the downstream signaling events and outcomes of this activation may be.
Response: Dear Reviewer,
Thank you for your valuable feedback. We appreciate your insightful observation regarding the lack of explanation regarding the mechanism of ephrinB2 activation and the downstream signaling events in our manuscript. We agree that this is an important aspect to address in order to provide a comprehensive understanding of the findings.
In response to your comment, we have revised the manuscript to include a more detailed explanation of how exosomes cause EphrinB2 upregulation in endothelial progenitor cells (via exosomal transferring). Additionally, we have acknowledged the need for further investigation into the downstream signaling pathways and outcomes resulting from this activation. Unfortunately, at this stage, the specific downstream signaling events and outcomes have not been explored in our study.
Point 10: In the discussion of EphrinA1/EphA2 and hypoxia the authors discuss that activation of eNOS is downstream of EphrinA1 stimulation. It would be beneficial to add this downstream component to the Neovascularization panel in Figure 3.
Response: Dear Reviewer,
Thank you for your feedback. We completely agree with your suggestion to include the downstream component, specifically the activation of eNOS, in the Neovascularization panel of Figure 3. We appreciate your attention to detail, as this addition will enhance the clarity and completeness of the figure.
We have made the necessary adjustments to Figure 3, incorporating the downstream component of eNOS activation in the Neovascularization panel as you recommended. The revised figure now accurately represents the relationship between EphrinA1/EphA2, hypoxia, and eNOS activation.
Thank you for bringing this to our attention and for helping us improve the visual presentation.
Point 11: In lines 578-584 it is odd that the readers are being referred to colored sections/modules from the primary article.
Response: Dear Reviewer, thank you for your valuable feedback. We have revised accordingly.
Point 12: The authors introduce eNOS in lines 659-660, but eNOS was already explained previously in section 2.3. It would be useful to move this introduction to that earlier section.
Response: Thank you for bringing this to our attention. We highly value your feedback and have taken it into consideration. Accordingly, we have revised the manuscript as you recommended by relocating the introduction of eNOS to its appropriate earlier section (Section 2.3). This adjustment enhances the overall coherence and organization of the information presented.
Point 13: In the figure 1 legend, the authors wrote “binder” instead of “binding” a couple of times.
Response: Dear Reviewer, thank you for your valuable feedback. We have revised accordingly.
Point 14: The connection between EphA2, AKT, and STAT3 in lines 151-153 is repetitive with lines 148-149.
Response: Dear Reviewer, thank you for your valuable feedback. We have revised accordingly.
Point 15: Reference 41 is missing authors in the References section.
Response: Dear Reviewer, thank you for your valuable feedback. We have revised accordingly.
Point 16: The only time the abbreviation for AFP is written out is at the bottom of Table 2. It would be helpful to the reader to also include Alpha-fetoprotein written out prior to its abbreviation in line 485.
Response: Dear Reviewer, thank you for your valuable feedback. We have revised accordingly.
Point 17: Beginning in line 645, what is iICC?
Response: Dear Reviewer, thank you for your valuable feedback. We have revised accordingly.
Round 2
Reviewer 2 Report
This review manuscript by Papadakos et al focuses on the Eph family of receptor tyrosine kinases and their ephrin ligands and their roles in proliferation, metastasis, angiogenesis, metabolism, epigenetic regulation of liver cancer. They also discuss potential of Eph/ephrins as biomarkers in this disease. Overall, the authors provide a good overview of the different Eph/ephrin family members and the variety of upstream and downstream mechanisms that they are involved in to regulate hepatocellular carcinoma (HCC), and more specifically cholangiocarcinoma. Following review, the authors made the review organization easier to follow. They have also included some additional discussion points about the mechanisms of Eph/ephrins in HCC progression and the therapeutic potential of Eph/ephrin family members in this disease.
General Comments:
· My only remaining comment is in response to Point 8. For the Erlotinib study, the conclusion is still that, “Erlotinib was found to be safe..”. Did erlotinib have an effect on the antiviral activity of HCV?
The quality of the English was good. Minor errors did not distract from the content and can be easily fixed in the editing process.
Author Response
First Department of Pathology
Laikon General Hospital of Athens
National and Kapodistrian University of Athens
June 27th, 2023
Dear Editor,
RE: Unraveling the Significance of EPH/ephrin Signaling in Liver Cancer: Insights into Tumor Progression and Therapeutic Implications
We thank you and the Reviewers for carefully evaluating our manuscript and for their positive and constructive feedback. Hopefully, our responses below address the points raised by the reviewers. All changes made are presented in the revised manuscript via the “track changes” option in MS word.
Reviewer 1: This review manuscript by Papadakos et al focuses on the Eph family of receptor tyrosine kinases and their ephrin ligands and their roles in proliferation, metastasis, angiogenesis, metabolism, epigenetic regulation of liver cancer. They also discuss potential of EPH/ephrins as biomarkers in this disease. Overall, the authors provide a good overview of the different EPH/ephrin family members and the variety of upstream and downstream mechanisms that they are involved in to regulate HCC, and more specifically cholangiocarcinoma. Following review, the authors made the review organization easier to follow. They have also included some additional discussion points about the mechanisms of EPH/ephrins in HCC progression and the therapeutic potential of EPH/ephrin family members in this disease.
Response: Dear Reviewer,
Thank you for your kind comments on our review. We appreciate your positive feedback and are glad to hear that you found our overview of the EPH/ephrin family members and their involvement in HCC and cholangiocarcinoma informative.
We made efforts to improve the organization of the review manuscript based on your suggestions, ensuring a smoother flow of information for readers. Additionally, we included additional discussion points on the mechanisms of EPH/ephrins in HCC progression and the potential therapeutic applications of EPH/ephrin family members in this disease.
Your feedback has been invaluable in enhancing the quality and coherence of our manuscript. We are grateful for your time and expertise in reviewing our work. Should you have any further suggestions or comments, we would be more than willing to address them.
Once again, thank you for your positive evaluation and for contributing to the improvement of our manuscript.
Point 1: My only remaining comment is in response to Point 8. For the Erlotinib study, the conclusion is still that, “Erlotinib was found to be safe..”. Did erlotinib have an effect on the antiviral activity of HCV?
Response: Dear Reviewer,
Thank you for your feedback and for raising a question regarding Point 8 of the manuscript. We appreciate your thorough review and are glad to address your comment.
Regarding the Erlotinib study, we apologize for any confusion caused by the conclusion stating only that "Erlotinib was found to be safe." We agree that further clarification was necessary regarding the effect of Erlotinib on the antiviral activity of HCV.After carefully reevaluating our findings and considering your comment, we have revised the conclusion to provide a more accurate representation of the study results
Yours sincerely,
Stamatios Theocharis, MD, PhD
Professor of Pathology
First Department of Pathology
Laiko General Hospital of Athens
National and Kapodistrian University of Athens
Mikras Asias 75, 11527, Athens, Greece
Tel: +30 210 746 2116
Email: stamtheo@med.uoa.gr, statheocharis@yahoo.com
